# WHEN THE GOLD ANSWER ISN'T THE ONLY RIGHT ONE: EVALUATING DATABASE QA VIA LLM-INDUCED RULE GUIDANCE

## ABSTRACT

Current NL2SQL evaluation relies heavily on execution accuracy (EX), which measures correctness by comparing query results against ground truth at the string level. While effective for traditional supervised models that produce uniform outputs, this metric proves inadequate in the LLM era, where diverse yet semantically equivalent SQL queries can correctly answer the same natural language question. To address this limitation, we investigate LLM-based evaluation for NL2SQL tasks and propose a rule generation-enhancing framework. It leverages a training dataset with annotated correctness labels through a three-step learning process: data clustering, intra-cluster rule summarization and refinement, and inter-cluster rule aggregation. So the model learn from labeled data through evaluation rule synthesis rather than parameter updates. The generated rules are integrated into LLM evaluation prompts during testing. We conduct experiments across three datasets, covering three evaluation scenarios: (1) identifying semantically correct predictions that differ in execution results from reference SQL, (2) distinguishing functionally different SQL queries that produce identical execution results, and (3) evaluating generated SQL correctness in the absence of reference queries. Our results demonstrate that traditional EX metrics show poor alignment with human annotations, while LLMs exhibit strong potential for this evaluation task. Our rule-generation framework consistently enhances LLMs' performance across all datasets and model variants. It effectively learns dataset-specific evaluation rules, and these learned rules can be successfully transferred to smaller models to improve their evaluation capabilities.

## 1 INTRODUCTION

Natural language to SQL (NL2SQL) transforms natural language queries into database queries based on specified database schemas, enabling accessible database analysis for all users. Recent advances in large language models (LLMs) have substantially improved NL2SQL performance, achieving notable results on benchmarks such as Spider (Yu et al., 2018) and BIRD (Li et al., 2023), thereby narrowing the gap between user intent and data analysis (Hong et al., 2024; Liu et al., 2024b; Gorti et al., 2024; Wang et al., 2025).

In NL2SQL research, the dominant evaluation metric is Execution Accuracy (EX), which determines correctness by comparing execution results using string matching. However, this metric suffers from two fundamental limitations. First, identical execution results to reference SQL do not necessarily indicate SQL correctness, as queries may coincidentally yield expected answers through erroneous logic. To address this issue, prior work has proposed constructing comprehensive test suites (Zhong et al., 2020), employing function-level verification models (Zhan et al., 2025), and leveraging LLM-based evaluation methods (Zhao et al., 2023). Second, SQL queries having different execution results with reference SQL may still be semantically correct if they faithfully capture the user's intent. This challenge parallels challenges in natural language generation tasks, such as machine translation and question answering, where diverse outputs can be equally valid and desirable.

Research addressing this second limitation has been notably scarce, primarily because public datasets such as Spider and BIRD encourage models trained on labeled data to produce uniform

outputs that closely match reference solutions, making semantically equivalent but syntactically different cases relatively uncommon. However, the rise of LLM-based approaches in practical applications has changed this landscape significantly. In the LLM era, task-specific training is often unnecessary, and satisfactory performance can be achieved through sophisticated prompt engineering alone. This paradigm shift introduces substantially greater diversity in system outputs, rendering the second challenge increasingly prevalent and critical for accurate evaluation.

For example, the question "Did institution B win the bid for bond A?" (shown in Figure 1) can be answered either by retrieving a specific winning bid record or by constructing a Boolean SQL query that returns Yes or No. Similarly, the query "What was the quotation situation yesterday?" may be interpreted through different combinations of fields, since the term situation is ambiguous and may refer to institution, bond name, bond type, quotation amount, quotation time, etc. In practice, users are often concerned with only a subset of these fields, making multiple field combinations equally valid. This diversity of correct outputs limits the ability of a single ground-truth SQL to represent all acceptable system responses. Traditional automatic evaluation methods based on SQL or execution results fail

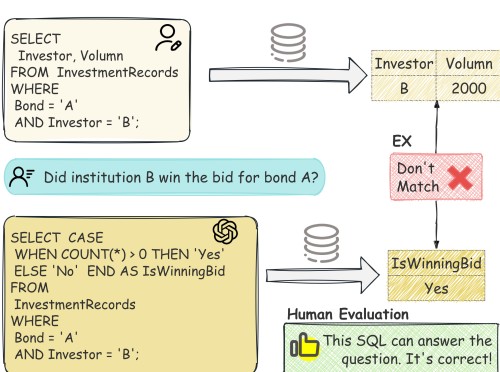

Figure 1: Generated SQL has different results with reference SQL, but correctly answers the question.

to capture such legitimate variations and frequently misclassify correct outputs as incorrect. Human evaluation, while capable of addressing this issue, is costly and inefficient, rendering it impractical for large-scale, iterative system development.

Therefore, we investigate the ability of LLMs in evaluating NL2SQL results and introduce HLSemEval (Human-Like Semantic Evaluation), a novel framework that simulates human judgment with LLMs. The framework first derives evaluation rules from labeled data by learning from human annotators' decisions on a small number of examples, and then applies these rules to assess whether model-generated SQL queries satisfy intended semantic and structural requirements.

Experiments across three complementary benchmarks highlight the effectiveness and generality of HLSemEval. On the Bond-QA dataset, which focuses on semantic correctness, it improves the evaluation F1 score of `DeepSeek-V3` from 92.40 to 96.13. On the Spider-Pair dataset, which focuses on functional equivalence, it achieves AUC scores competitive with the state-of-the-art FuncEval-GMN while using less than 10% of its training data, underscoring strong data efficiency and scalability. On the NL2SQL-BUGs dataset, which focuses on semantic correctness without reference SQL, it raises `DeepSeek-V3` F1 by 6.94 points, showing that reliable semantic rules can be distilled even without reference SQLs. Together, these results demonstrate that HLSemEval offers a practical, generalizable, and semantically faithful framework for NL2SQL evaluation.

## 2 RELATED WORKS

### 2.1 EVALUATION OF NL2SQL

NL2SQL evaluation concerns the correctness of SQL generated from natural language. Mainstream approaches include match-based metrics, graph-based methods, and LLM-based evaluators.

**Match-Based Evaluation** Popular benchmarks such as Spider and BIRD adopt evaluation schemes based on SQL matching and execution results, including Execution Accuracy (EX), Component Matching (CM), and Exact Match (EM). EX may yield false positives due to coincidental data distributions. CM and EM check structure but can miss valid semantic variants. To mitigate these issues, Zhong et al. (2020) proposes Test Suite Accuracy, which compares outputs across multiple constructed databases to approximate semantic equivalence. SQLSolver (Ding et al., 2023) addresses unbounded summations via the extended Linear Integer Arithmetic with Stars (LIA*) theory, enabling principled handling of nested, parameterized, and non-linear unbounded summations.

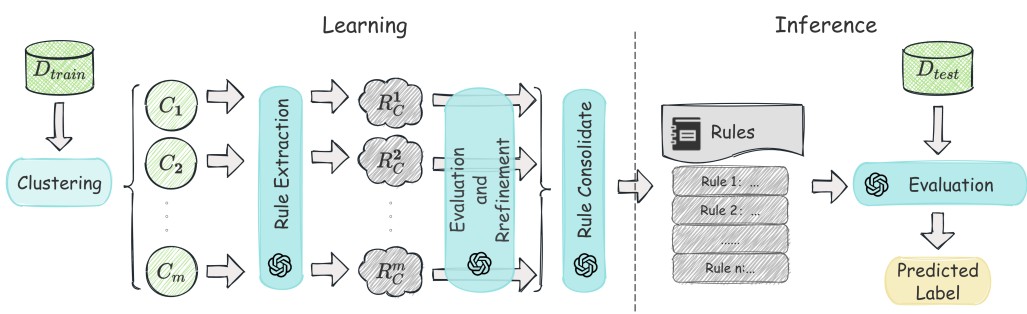

Figure 2: Overview of the Two-Stage Evaluation Framework: Rule Extraction (Learning) and Rule-Guided Evaluation (Inference).

**Graph-Based Evaluation**   Graph-based methods represent SQL as graphs and compare them via graph matching. Köberlein et al. (2024) model queries as nodes and define edit operations weighted by semantic differences; a shortest-path search yields a semantic distance. FuncEvalGMN (Zhan et al., 2025) encodes queries as Relation Operation Trees, builds program graphs integrating data and logic flows, and applies a Graph Matching Network to assess similarity. These methods abstract execution logic, tolerate syntactic differences, and judge equivalence through functional semantics.

**LLM-Based Evaluation**   LLMs are increasingly used as evaluators for generation tasks like question answering (Gu et al., 2024; Liu et al., 2024c;d; Wang et al., 2023). Some work uses Embedding-based methods, such as CodeBERTScore (Zhou et al., 2023) and CodeScore (Dong et al., 2025), which combine contextual representations with execution semantics to improve robustness. Other work employs LLMs directly. LLM-SQL-Solver (Zhao et al., 2023) uses two prompting strategies: "Miniature and Mull" for strict functional equivalence, and "Explain and Compare" for functional equivalence. FLEX (Kim et al., 2025) employs prompts that, when two SQL queries produce identical execution results, further inspect their logical structures to verify true equivalence, ensuring that the predicted query is not only result-equivalent but also logically faithful to the user question. When results diverge, it assesses whether the discrepancies remain acceptable in the task context. Despite these advances, the evaluation paradigm largely reduces to testing SQL functional equivalence, which remains difficult when a single question admits multiple valid answers (see Figure 1).

Overall, most automated NL2SQL evaluations center on SQL functional equivalence. This focus limits effectiveness when multiple answers are valid, can misalign with user intent, and slows NL2SQL system development. We address these limitations with *HLSemEval*, an LLM-based framework that learns human-annotated rules to handle multi-answer scenarios, improving semantic coverage, adaptability, and evaluation consistency.

## 3 METHODS

This section describes the design of *HLSemEval*, as illustrated in Figure 2. In the rule extraction phase (learning by extracting rules as opposed to learning by updating parameters), the system analyzes a set of human-annotated samples (i.e., the training set) to systematically derive evaluation rules that emulate human judgment. In the rule-guided evaluation phase (inference), these extracted rules are applied to guide an LLM in assessing SQL queries, enabling human-like evaluation.

### 3.1 PROBLEM DEFINITION

We consider the task of assessing whether a predicted SQL query $\hat{s}$ answers the user's question $q$. Specifically, it takes as input a tuple $(q, \hat{s}, s, \hat{e}, e)$, where

- $q$ is the natural language question posed by the user;
- $\hat{s}$ and $s$ are the predicted and reference SQLs, respectively (we use "reference SQL" rather than "gold SQL" to reflect the potential multiplicity of valid answers);

- $\hat{e}$ and $e$ are the execution results of $\hat{s}$ and $s$ on the target database schema $\mathcal{S}$.

The goal is to determine whether the predicted SQL query $\hat{s}$ satisfies the semantic intent of $q$ in the given context. The output is a binary correctness label $\hat{y} \in \{0, 1\}$, where 1 indicates semantic correctness and 0 otherwise.

## 3.2 LLM-BASED EVALUATION

Existing research has shown that the use of a multi-dimensional rubric to guide LLM evaluation and scoring improves the consistency and interpretability of results in comparison with human reviewers, as demonstrated by (Hashemi et al., 2024). Additionally, Pathak et al. (2025) highlights that task-specific rubrics, rather than generic ones, significantly improve the evaluation performance.

To emulate human judgment in the evaluation process, we adopt an LLM-based approach guided by a set of interpretable rules $\mathcal{R} = \{r_1, r_2, \ldots, r_M\}$, where each rule $r_i$ is formulated in natural language to enhance readability and interpretability. This process is formally defined as:

$$f_{\text{eval}}(\cdot) = \text{LLM}\big(\mathcal{R};\, q,\, \hat{s},\, s,\, \hat{e},\, e\big),$$

which produces a predicted correctness label $\hat{y}$ indicating whether the generated SQL meets the semantic intent of the input question under the given context.

Here, constructing an effective $\mathcal{R}$ is crucial for the accuracy of the evaluation. Similar to manually constructed rule knowledge, we believe that $\mathcal{R}$ for a task should be derived by analyzing and summarizing an existing dataset. Therefore, we assume the availability of a human-annotated dataset from which the rules can be learned.

$$\mathcal{D} = \big\{(q_j,\, \hat{s}_j,\, s_j,\, \hat{e}_j,\, e_j,\, y_j)\big\}_{j=1}^{N},$$

where $y_j \in \{0, 1\}$ denotes the human-provided correctness label for each instance. The objective is to extract a rule set $\mathcal{R}$ from this dataset that enables accurate prediction of $\hat{y}_j$:

$$\min_{\mathcal{R}} \sum_{i=1}^{N} (f_{\text{eval}}(\mathcal{R};\, q_j, \hat{s}_j, s_j, \hat{e}_j, e_j) - y_i)^2.$$

## 3.3 LEARNING BY EXTRACTING RULES

During the learning phase, we aim to leverage LLMs to automatically summarize the implicit decision logic embedded in human annotations, thereby extracting a rule set $\mathcal{R} = \{r_1, r_2, \ldots, r_M\}$ which is readable. This rule set is used during LLM-based evaluation. Following standard machine learning practice, we divide the original dataset into a training set $D_{\text{train}}$ and a test set $D_{\text{test}}$.

The learning process is shown in Figure 2. Initially, the rule set is empty $R \leftarrow \emptyset$, and all training samples are placed into a working set $D_{\text{train}}$. We then proceed with a clustering-based rule extraction pipeline as follows.

**Clustering.** Each input question $q$ in $D_{\text{train}}$ is projected into a vector space using an embedding model. Then, they are grouped into a set of clusters $C = \{c_1, c_2, \ldots, c_m\}$, to aggregate semantically similar questions. Semantically similar questions often correspond to similar SQL structures and database operations; placing them within the same cluster allows for fine-grained comparison, which helps the LLM discover subtle and precise evaluation rules.

**Rule Extraction.** For each semantic cluster $c \in C$, we perform rule extraction and optimization. The input samples $D_c = \{(x_i, y_i)\}_{i=1}^{N_c}$ of cluster $c$ are first formatted into (input, label) pairs. These formatted samples are then incorporated into a rule-extraction prompt, which is submitted to the LLM to generate the initial rule set $\mathcal{R}_c^{(0)}$ for cluster $c$.

**Self-Evaluation and Refinement.** Then, we assess $\mathcal{R}_c^{(0)}$ by using them to evaluate samples within cluster $c$ and computing the evaluation accuracy $a$. Define a threshold $\theta$, if $a < \theta$, we enter an iterative refinement phase indexed by $t = 1, 2, \ldots, T$ until $a \geq \theta$ or $t = T$.

In each refinement iteration, inspired by Boosting-style hard example reweighting, we emphasize the misclassified samples within the cluster and supplement them with a fixed proportion of correctly

classified samples to refine the rules. This strategy aims to enhance the rule coverage over "blind spot" samples that are not well captured by the current rules.

The selected samples $\mathcal{E}^{(t-1)}$ and the current rule set $\mathcal{R}_c^{(t-1)}$ are used to construct the prompt $P_{\text{refine}}^{(t)}$, where harder cases are given more emphasis by increasing their proportion within the samples used for rule refinement and explicitly instructing the model in the refinement prompt to focus on these cases. This refined prompt is then passed to the LLM to generate an updated rule set $\mathcal{R}_c^{(t)}$.

**Rule Consolidation.** Once rule extraction for all clusters is complete (in parallel), the individual rule sets $\mathcal{R}_c$ are consolidated via an LLM into a unified, merged, and deduplicated global set $\mathcal{R}$. This aggregation enhances global coverage while preserving the discriminative power of cluster-specific rules, and the resulting $\mathcal{R}$ is used for inference.

## 4 EXPERIMENTS

Our experiments are conducted on three datasets: **Bond-QA**, **Spider-Pair** (Zhan et al., 2025), and **NL2SQL-BUGs** (Liu et al., 2025), to cover three critical evaluation conditions: (i) predicted queries producing different execution results yet representing semantically correct SQL (Bond-QA), (ii) predicted queries producing the same execution results despite being functional inequivalence with reference SQL (Spider-Pair), and (iii) fine-grained *semantic error detection*, where only an $(q, \hat{s}, \mathcal{S})$ triple is provided without reference SQL (NL2SQL-BUGs).

Our experiments demonstrate that widely used metrics such as execution accuracy (EX) exhibit clear limitations, while LLMs show strong potential on this task. The proposed rule extraction method automatically learns task-specific evaluation rules from annotated data, adapting to different datasets and thereby improving model performance.

### 4.1 EXPERIMENTAL SETUP

**Bond-QA.** We collect 900 real-world DBQA samples from fixed-income workflows by in-house experts. We use 150 instances for rule learning and 750 for testing. The evaluator outputs a binary decision on whether the predicted SQL answers the question. Baselines: Miniature and Mull (Zhao et al., 2023)[1] and FLEX (Kim et al., 2025)[2]. FuncEvalGMN (Zhan et al., 2025) is omitted because it targets functional (not pragmatic) equivalence, but this dataset focuses on functionally different SQLs that both answer the question.

**Spider-Pair.** Zhan et al. (2025) construct the dataset using methods like LLM-based techniques on the Spider dataset, among others, evaluates EX judgments for cases where the results are identical but may be functionally not equivalent to the reference SQL. Unlike Bond-QA, this dataset investigates whether queries with structural differences but functionally equivalent outputs can be detected. We simulate limited-data training with 1,600 randomly selected samples (mixing functionally equivalent and non-equivalent pairs) which represents 10% of the training set used in the FuncEvalGMN method. We evaluate on the four official test sets (Spider, WikiSQL, BIRD, and Spider-DK). Inputs consist only of the SQL pair $(\hat{s}_j, s_j)$ (no execution results as they are the same). Following (Zhan et al., 2025), we use the AUC metric for evaluation, and we modify the evaluator to output a score in $[0, 1]$, indicating functional consistency.

**NL2SQL-BUGs.** Semantic error detection over $(q, \hat{s}, \mathcal{S})$ (Liu et al., 2025) without reference SQL. We use binary labels only, splitting 505/1,513 for train/test. To prevent leakage, we cluster by database schema: samples from the same schema are assigned to a cluster.

We use `text-embedding-3-small` as the embedding model, with temperature 0 for all LLM calls, and use KMeans for clustering. Average cluster sizes are 15 for Bond-QA and 60 for Spider-Pair. Further implementation details are provided in Appendix A.1.

---

[1] https://github.com/ZhaoFuheng/LLM-SQL-Solver
[2] https://github.com/HeegyuKim/FLEX

## 4.2 METRICS

For **Bond-QA** and **NL2SQL-BUGs**, we evaluate the alignment between predicted labels and gold labels using standard classification metrics: Accuracy, Precision, Recall, and F1. For the **Spider-Pair** dataset, we follow the evaluation protocol of FuncEvalGMN (Zhan et al., 2025) and report the Area Under the ROC Curve (AUC). We report classification metrics as percentages (0–100), while AUC is reported on the [0,1] scale; improvements are expressed in percentage points (pts).

## 4.3 RESULTS AND ANALYSIS

### 4.3.1 RESULTS ON BOND-QA

Table 1: Evaluation results on the Bond-QA dataset. "–" denotes not applicable.

| Method | No Rules | | | | + Rules | | | | |
|---|---|---|---|---|---|---|---|---|---|
| | Acc. | Prec. | Rec. | F1 | Acc. | Prec. | Rec. | F1 | ΔF1 |
| EX | 48.89 | 99.41 | 42.32 | 59.36 | – | – | – | – | – |
| Miniature & Mull | 14.82 | **100.00** | 9.38 | 17.14 | – | – | – | – | – |
| Flex | 89.44 | 97.56 | 91.04 | 94.19 | – | – | – | – | – |
| DeepSeek-V3 | 92.40 | 98.93 | 92.84 | 95.79 | 96.13 | 97.47 | 98.43 | 97.95 | +2.16 |
| Kimi-K2 | 94.67 | 96.77 | 97.59 | 97.18 | 96.67 | 96.96 | **99.57** | 98.25 | +1.07 |
| GPT-4.1 | 95.07 | 98.27 | 96.45 | 97.35 | **96.80** | 98.85 | 97.73 | **98.29** | +0.94 |
| Qwen3-32B | 91.87 | 98.64 | 92.62 | 95.54 | 91.07 | 99.23 | 91.21 | 95.05 | -0.49 |
| Qwen3-32B (+ds) | – | – | – | – | 94.27 | 98.96 | 94.89 | 96.89 | +1.35 |

Table 1 summarizes the overall performance of various evaluation methods on the Bond-QA dataset. Overall, we observe clear differences in effectiveness across methods. In particular, the results reveal four key aspects that explain the observed performance patterns:

**Multiple valid results are common in the real world, and the EX metric is ineffective.** EX attains extremely high precision (99.41), correctly rejecting nearly all genuinely incorrect queries. However, this comes at the expense of low recall (42.32), because it labels as incorrect any semantically correct variant that does not exactly match the single reference output. This precision–recall imbalance exposes a structural limitation of execution-only evaluation: it under-covers the legitimate diversity of correct answers common in real-world workloads.

**Carefully crafted manual prompts may fail to generalize across datasets.** Different evaluation objectives demand different prompt strategies, leading to strikingly varied outcomes. For instance, under the `DeepSeek-V3` setting, **Miniature and Mull**, while attaining perfect precision (100.00), suffers from extremely poor overall accuracy (14.82) and recall (9.38). This approach merely presents the LLM with two SQLs and a schema, asking it to infer equivalence under hypothetical execution. Lacking actual execution results or contextual grounding, the LLM struggles with realistic database complexity, resulting in severe under-detection of correct queries. In contrast, **FLEX** employs a more adaptive design that integrates execution evidence and external knowledge, enabling it to capture subtler semantic nuances and achieve much higher accuracy (89.44) and F1 (94.19). Nevertheless, FLEX still lags behind the best-performing methods on Bond-QA, since its correctness criterion emphasizes SQL consistency rather than the true evaluation goal: determining whether the SQL answers the user's question regardless of structural or executional variations. When prompts are explicitly aligned with the Bond-QA objective, strong performance can already be obtained even without additional rule guidance ("no rules" setting).

**Automatically extracted rules provide further gains.** Integrating rules consistently enhances results across models. For instance, accuracy improves from 92.40 to 96.13 for `Deepseek-V3` (Liu et al., 2024a), from 94.67 to 96.67 for `Kimi-K2` (Moonshot AI, 2024), and from 95.07 to 96.80 for `GPT-4.1`. These gains demonstrate that rules distilled from labeled data capture subtle semantic differences that instruction-only setups may miss.

**Extracted rules from strong models can be applied to weak models.** When `Qwen3-32B` is used as the rule extractor, downstream performance decreases rather than improves, indicating limited capacity to abstract reliable, task-level rules. In contrast, rules distilled by a stronger model such as `Deepseek-V3` yield clear gains even when applied to `Qwen3-32B` at evaluation time (see the "Qwen3-32B (+ds)" row). This contrast shows that while rule guidance is broadly beneficial, extracting high-quality rules is capacity-sensitive. Once the rule set is established, however, it transfers effectively to smaller models.

### 4.3.2 RESULTS ON SPIDER-PAIR

Table 2: AUC results on Spider-pair dev sets. **SP** = Spider-Pair-dev, **BP** = BIRD-Pair-dev, **SDP** = Spider-DK-Pair-dev, **WP** = WikiSQL-Pair-dev.

| Method | SP | BP | SDP | WP | Avg. |
|---|---|---|---|---|---|
| G-eval (GPT-4) | 0.6386 | 0.7042 | 0.7212 | 0.6139 | 0.6695 |
| Test Suite | 0.9637 | 0.9267 | 0.9277 | – | 0.9394 |
| GPT-4.1 | 0.9325 | 0.9511 | 0.9586 | 0.9432 | 0.9464 |
| Ours (GPT-4.1) | 0.9559 | **0.9497** | 0.9704 | 0.9769 | 0.9632 |
| FuncEvalGMN | **0.9750** | 0.9272 | **0.9753** | **0.9910** | **0.9671** |

Table 2 reports the AUC on four development sets. For G-eval (GPT-4), Test Suite, and FuncEval-GMN, we directly adopt the results reported in the FuncEvalGMN paper to ensure fair comparability. Since version differences of LLMs may lead to non-trivial variance, we consistently use the paper-reported values rather than re-running these methods.

**Execution/string-based evaluators have inherent limits despite strong AUC.** Spider-Pair exposes a key failure mode of EX-style evaluation: even when $\hat{e} = e$, two queries can implement different semantics, and EX will mark them as correct simply because their outputs coincide by accident. Test-suite evaluators can achieve high AUC (0.9394 across the three released sets), but they depend on building task- and schema-specific sub-suites, and require database access for denotation checks. This combination of schema customization and database dependency limits their portability and general applicability. By contrast, LLM-based evaluators (e.g., GPT-4.1 and ours) assess functional equivalence directly from the SQLs without execution, and more reliably flag nonequivalence that arises from structural or logical disparities rather than surface similarity or coincidental result ties, achieving consistently higher AUC scores across diverse evaluation sets.

**Automatically extracted rules provide consistent gains.** Compared with GPT-4.1 without rule extraction, our method improves AUC by 2.34 pts on Spider-Pair-dev, 1.18 pts on Spider-DK-Pair-dev, and 3.37 pts on WikiSQL-Pair-dev, with a marginal decrease of 0.14 pts on BIRD-Pair-dev. A small decline on BIRD is reasonable because BIRD exhibits greater schema and SQL complexity than Spider, so rules distilled mainly from Spider data are not fully general and can at times constrain judgments in BIRD-specific situations. Nevertheless, the overall effect of adding rules remains positive, as reflected by the average AUC gain of 1.68 pts across development sets.

**Rules distilled by stronger models transfer effectively to weaker ones.** On Spider-Pair-dev, we also evaluated a smaller `Qwen3-32B` evaluator and found that rule guidance helps: it attains AUC 0.8869 vs. 0.8467 without rules (+4.02 pts). Moreover, when `Qwen3-32B` is guided by rules distilled by `GPT-4.1`, its AUC rises to 0.9152, surpassing its self-extracted-rules variant (0.8869; +2.83 pts) and the no-rule baseline (0.8467; +6.85 pts). These results corroborate our earlier observation: rule guidance is broadly beneficial, but extracting high quality rules requires sufficient model capacity—rules distilled by stronger models yield larger gains.

**Rule-guided LLM evaluators approach specialized systems with far less supervision.** FuncEvalGMN (Zhan et al., 2025) is a specialized evaluator that models SQL queries as program graphs and leverages a Graph Matching Network to measure functional equivalence. With less than 10% of its training data, our method still achieves near-parity AUC (avg. 0.9632 vs. 0.9671), offering comparable accuracy at a fraction of the supervision cost and with simpler deployment.

### 4.3.3 RESULTS ON NL2SQL-BUGS

Table 3: Results on NL2SQL-BUGs dataset.

| Model / Version | Acc. | Prec. | Rec. | F1 |
|---|---|---|---|---|
| DeepSeek-V3 (no rules) | 73.10 | **86.21** | 56.92 | 68.57 |
| DeepSeek-V3 (rules) | 76.80 | 82.85 | 69.36 | 75.51 |
| GPT-4.1 (no rules) | 78.39 | 84.79 | 70.77 | 77.15 |
| GPT-4.1 (rules) | **79.64** | 85.65 | **72.69** | **78.64** |

On the NL2SQL-BUGs benchmark (Table 3), we do not compare with execution-based methods such as EX since this dataset lacks reference SQLs. Instead, we demonstrate that LLMs can still derive reliable rules through analysis and summarization, enabling correctness judgment without reference SQL. Applying our rule-guided method improves the accuracy of `DeepSeekV3` by 3.70 pts and the F1 score by 6.94 pts, and improves the accuracy of `GPT-4.1` by 1.25 pts and the F1 score by 1.49 pts. These results demonstrate that even with supervision restricted to $(q, \hat{s}, \mathcal{S}, y)$, the evaluator can still acquire generalizable and semantically meaningful rules that expand correct coverage and improve end performance, highlighting our method's robustness and extensibility.

## 4.4 CASE STUDY

We illustrate through several examples that the rules extracted by the model reflect dataset-specific task settings. For instance, in the Bond-QA dataset, the rules mainly explain the circumstances under which different execution results are acceptable, while in the Spider-Pair dataset, the rules focus on situations where functional discrepancies are unacceptable. These rules are difficult to fully write out directly and need to be derived through analysis and summarization of the dataset. In this section, we discuss a single Bond-QA case study, with additional examples provided in Appendix A.3.

Consider the question: "How many secondary-market cash-bond trades and repo trades with A occurred in 2024, reported as separate counts?" (see Appendix A.3.1). The predicted SQL groups trades into Cash-Bond, Repo and returns one row per category, while the reference SQL uses `SUM(CASE...)` to output two columns. Both constrain to counterparty A, the year 2024, and confirmed trades.

Without rules, the prediction is marked incorrect as repo trades are zero because the grouped result omits the "Repo" row, and the LLM considers this as an incomplete answer instead of inferring zeros.

Our extracted rule states: *When predicted and reference queries differ in aggregation form, grouping fields, filters, or naming, but operate at the same granularity and filtering logic required by the question, and the requested values can be directly inferred—including treating a missing group as zero—classify as correct.*

Applying this rule, the prediction satisfies all core constraints and implies `repo = 0`. Hence, the predicted SQL is judged correct. This case shows how rule guidance recovers valid answers that instruction-only LLMs fail to recognize and would otherwise label as false negatives.

## 4.5 ABLATION STUDY

Table 4: Ablation study on components.

| Setting | Acc. | Prec. | Rec. | F1 |
|---|---|---|---|---|
| Full | **85.50** | **83.50** | **97.73** | **90.05** |
| w/o Cluster | 78.63 | 77.78 | 95.45 | 85.71 |
| w/o Consolidation | 80.16 | 78.70 | 96.59 | 86.73 |

### 4.5.1 COMPONENT-LEVEL ABLATION

To assess the contribution of each component to evaluation performance, we built a focused ablation test set of 131 Bond-QA samples with large execution-result discrepancies, representing the dataset's most challenging cases. Instead of covering the full dataset, this compact but representative subset isolates scenarios where execution matching fails. Unlike cases with consistent outputs, where correctness is trivial to verify, these samples require the evaluator to emulate human reasoning and apply rule-guided semantic judgment to both SQL and execution results.

Table 4 reports results using `DeepSeek-V3` as the base model. The complete algorithm—retaining both question clustering and rule consolidation—achieves the best overall performance and serves as the reference baseline. Removing the clustering step (w/o Cluster), where samples are randomly divided into groups to control input length, reduces accuracy to 78.63 and F1 to 85.71. This shows that clustering aligns semantically related samples, facilitating high-quality rule extraction. Similarly, removing rule consolidation and keeping only local cluster rules (w/o Consolidation) lowers accuracy and F1 to 80.16 and 86.73. This highlights the consolidation module's role in resolving inter-cluster conflicts and ensuring global evaluation consistency.

### 4.5.2 TRAINING SIZE ABLATION

This section investigates the impact of training sample size on the quality of extracted rules and the overall evaluation performance, measured by AUC. Using the Spider-pair-dev dataset as the benchmark, we vary the size of the training set used for rule extraction under controlled conditions to examine its influence.

As shown in Fig. 3, even with only 200 annotated examples, rule-guided evaluation achieves a notable improvement over the no-rules baseline: the AUC rises from 0.9325 to 0.9442. This demonstrates that a small amount of supervision can already enhance the model's capacity to judge semantic equivalence between SQLs. When the training size increases from 200 to 400 samples, the AUC

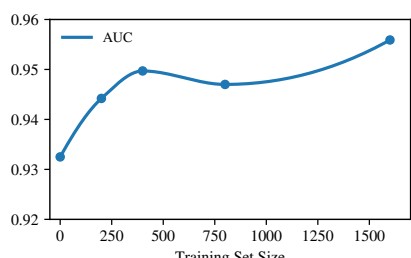

Figure 3: Performance with different training set sizes (Spider-pair-dev).

steadily improves from 0.9442 to 0.9497, reflecting better rule generalization and broader coverage of subtle semantic patterns. Although a slight dip is observed at 800 samples (0.9470), further expansion to 1,600 samples leads to the highest AUC of 0.9559. This pattern suggests a generally positive relationship between training size and evaluation performance, but with diminishing returns once the rule set becomes sufficiently comprehensive.

In practice, our rule-driven evaluation framework can achieve strong performance without massive training data: a moderate number of representative and diverse examples is sufficient to approach optimal accuracy. Beyond this point, focusing on improving the quality and diversity of training samples may be more effective than simply increasing quantity.

## 5 CONCLUSION AND DISCUSSION

We propose HLSemEval, an LLM-based evaluation framework that leverages LLMs to address the challenge of evaluating multiple valid execution results in NL2SQL, where existing automatic methods fail to recognize semantically equivalent variants. By inducing interpretable rules from human-labeled data and guiding LLM reasoning with these rules, HLSemEval achieves low-resource, interpretable, and scalable automatic evaluation.

HLSemEval can be improved from multiple directions, such as refining clustering methods or adopting multi-round iterations over the overall workflow to compare and adjust samples and rules across clusters. Nevertheless, our results demonstrate that systematically summarizing rules from datasets can consistently enhance large models' performance on this task, making HLSemEval a promising approach.

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

# A APPENDIX

## A.1 DATASETS AND EXPERIMENTAL SETUP (DETAILS)

**Bond-QA.** **Bond-QA** contains 900 real-world DBQA samples from an operational fixed-income system spanning issuance, inquiry, investment, and trading. Domain experts authored NL questions; technical staff paired reference SQLs. Predicted SQLs were produced by an existing workflow and manually annotated for semantic correctness against the references. Unlike strict-equivalence datasets, Bond-QA judges whether a generated SQL can validly answer the question even if its execution result differs, reflecting the fact that multiple semantically correct SQLs may exist. In our experiments, we use 150 instances for rule learning and 750 for testing. The evaluator outputs a binary judgment on whether the predicted SQL answers the question. We compare to Miniature and Mull (Zhao et al., 2023)[3] and FLEX (Kim et al., 2025)[4], and exclude FuncEvalGMN because it emphasizes functional rather than pragmatic equivalence.

**Spider-Pair.** **Spider-Pair** (Zhan et al., 2025) is derived from Spider and enforces strict SQL equivalence. Each sample includes a gold SQL, a predicted SQL, and a correctness label, with training and test domains disjoint to avoid leakage. The corpus provides roughly 18,000 training samples and four test sets (Spider, WikiSQL, BIRD, Spider-DK), reducing false positives where structurally different SQLs coincidentally yield identical results. In our evaluation, we simulate limited-data settings via a curated set of 1,600 training samples combining: (i) false positives (identical execution

---

[3] https://github.com/ZhaoFuheng/LLM-SQL-Solver
[4] https://github.com/HeegyuKim/FLEX

results yet labeled incorrect) and (ii) correctly labeled pairs to prevent overfitting. Because Spider-Pair lacks natural language questions, our inputs use only the SQL pair $(\hat{s}_j, Y_j)$ (without execution results). The evaluator outputs a real-valued score in $[0, 1]$ indicating functional consistency.

**NL2SQL-BUGs.** NL2SQL-BUGs (Liu et al., 2025) targets semantic error detection. Each instance provides an NL question, a DB schema, and a candidate SQL; the task is to decide whether the SQL semantically matches the NL over the given DB. The benchmark has 2,018 expert-annotated instances across 9 main and 31 sub-categories, including correct and incorrect cases with detailed error labels. In our experiments, we *only* use the binary correctness label: 505 samples (25%) for training and 1,513 (75%) for evaluation. Different from the original question-similarity clustering, we cluster by database schema so that queries from the same schema do not cross splits, enforcing stricter generalization.

---

**Algorithm 1** LLM-based Rule Extraction and Refinement in a Cluster

---

**Require:** Cluster $c$ with $N_c$ samples
**Ensure:** Refined rule set $R_c$
 1: {**Phase 1: Initial Rule Extraction using LLM**}
 2: $P_{\text{extract}} \leftarrow$ BuildExtractionPrompt(Template$_{\text{extract}}$, $c$)
 3: $R_0 \leftarrow$ **LLM_Extract**($P_{\text{extract}}$)
 4: {**Phase 2: Iterative Refinement using LLM**}
 5: **for** $j = 1$ to $T_{\text{refine}}$ **do**
 6:     $E_{\text{results}} \leftarrow \emptyset$
 7:     **for all** $s_c \in c$ **do**
 8:         $\hat{y} \leftarrow$ **Inference**($R_{j-1}, s_c$)
 9:         $E_{\text{results}} \leftarrow E_{\text{results}} \cup \{\hat{y}\}$
10:     **end for**
11:     $acc_j \leftarrow$ CalculateAccuracy($E_{\text{results}}, c$)
12:     **if** $acc_j \geq \theta$ **then**
13:         **break**
14:     **end if**
15:     {Refine rules with error cases}
16:     $A_{\text{refine}} \leftarrow$ SelectRefinementSamples($E_{\text{results}}, c$)     // Includes misclassified and a subset of correct samples
17:     $P_{\text{refine}} \leftarrow$ BuildRefinementPrompt($R_{j-1}, A_{\text{refine}}$)   // Constructed from the selected samples
18:     $R_j \leftarrow$ **LLM_Refine**($P_{\text{refine}}$)
19: **end for**
20: **return** $R_{\arg\max_j acc_j}$

---

To provide a clearer understanding of our pipeline, Algorithm 2 illustrates the full procedure of rule-guided assessment using LLMs

---

**Algorithm 2** Rule-guided evauation method with LLM

---

**Require:** Training set $D_{train}$, Test set $D_{test}$
**Ensure:** Final rule library $R$
 1: Initialize rule library $R \leftarrow \emptyset$
 2: Initialize working set $D'_{\text{train}} \leftarrow D_{\text{train}}$
 3: {**Step 1: Semantic Clustering**}
 4: $C \leftarrow$ SemanticClustering($D'_{\text{train}}$) {e.g., K-means over question embeddings}
 5: {**Step 2: Cluster-wise Rule Extraction**}
 6: **for all** $c \in C$ **do**
 7:   $R_c \leftarrow$ **ExtractAndRefineRules**($c$)
 8:   $R \leftarrow R \cup R_c$
 9: **end for**
10: {**Step 3: Rule Aggregation Module**}
11: $R \leftarrow$ MergeRules($\{R_c\}_{c \in C}$)
12: {**Step 4: Final Evaluation on Test Set**}
13: $acc_{\text{test}} \leftarrow$ Evaluate($R, D_{\text{test}}$)
14: **return** $R$

---

## A.2 PROMPT TEMPLATES USED IN EXPERIMENTS

For each dataset, we designed four types of prompt templates to support different stages of our rule-based evaluation framework:

- **Rule Extraction** – to extract candidate evaluation rules from learning sets.
- **Rule Refinement** – to refine extracted rules.
- **Rule Aggregation** – to consolidate refined rules into a compact, generalizable rule library.
- **Evaluation** – to assess whether a predicted SQL is correct.

In the **Evaluation** stage, the overall instruction remains identical across conditions; the only difference lies in the presence or absence of the extracted rules in the input. This allows us to directly quantify the impact of rule guidance while keeping other evaluation factors constant.

### A.2.1 PROMPT TEMPLATES USED IN SPIDER-PAIR DATASET

**Prompt Template: Rule Extraction**

---

**Task Description.** We are building an LLM-based SQL equivalence evaluation system to determine whether a predicted SQL query (`pred_sql`) is *functionally equivalent* to the ground-truth SQL query (`gold_sql`).
The complete evaluation process consists of two phases:

1. **Rule Extraction (Training)**: Derive evaluation rules from labeled samples to simulate the logic and criteria used by human annotators.

2. **Rule-Based Evaluation (Inference)**: Use the extracted rules to determine whether a predicted SQL query is functionally equivalent to the ground-truth SQL.

You are now responsible for the **Rule Extraction** phase.

**Input.**

- A set of labeled samples, each containing:
  - Predicted SQL
  - Ground-truth SQL
  - Manual label (equivalent / non-equivalent)
- Input sample overview: `{samples_info}`

**Rule Extraction Guidelines.**

- Compare the structural and semantic differences between predicted SQL and ground-truth SQL.
- Learn the reasoning behind human annotations (equivalent / non-equivalent).
- Extract rules that explain under what structural/semantic conditions a predicted SQL is (or is not) functionally equivalent to a ground-truth SQL.

**Requirements for Each Rule.**

- Each rule must capture a specific structural or semantic difference in SQL that directly causes equivalence or non-equivalence.
- The rule must **not** rely on execution results or result sets.
- Rules should be both **specific** and **generalizable**:
  - Specific: Describe the logical structure and semantic pattern clearly.
  - Generalizable: Apply to a class of SQLs beyond a single example.
- Example snippets may be used to illustrate a rule but must not form the basis of the rule.
- Be rigorous when defining equivalence; avoid overgeneralized rules that may result in false positives.

---

**Output Format.**

- First, provide your reasoning process step-by-step.
- Then output the final rules in the following strict JSON format:

```json
{
  "rules": [
    {
      "description": "A clear logical explanation of why a certain
    pattern leads to an equivalent or non-equivalent judgment.",
      "type": "True/False" // True = equivalent, False = non-
    equivalent
    }
  ]
}
```

**Prompt Template: Rule Refinement**

**Task Description.** We are building an LLM-based SQL equivalence evaluation system to judge whether a predicted SQL query (`pred_sql`) is *functionally equivalent* to a ground-truth SQL query (`gold_sql`).
The complete evaluation process includes two phases:

1. **Rule Extraction (Training)** – Deriving evaluation rules from labeled samples to simulate the logic and standards of manual evaluation (labels).

2. **Rule-Based Evaluation (Inference)** – Using the extracted rules to evaluate whether a predicted SQL is functionally equivalent to a ground-truth SQL.

You are now in the iterative refinement part of the **Rule Extraction** phase. You need to modify and improve the existing rules based on the provided samples, ensuring that the revised rules can be correctly applied to the current set of samples.
You will receive a set of current evaluation rules and a set of samples that were evaluated using these rules. Your task is to analyze the misjudgments in these samples, and modify or add rules to more accurately simulate human evaluation logic, ensuring the revised rules work correctly on this set of samples.

**Input.**

- **Current Rule Set:** Generated in the previous phase, containing multiple rules for judging functional equivalence.
- **Error Sample Set:** Each sample includes the predicted SQL, ground-truth SQL, the manual label, and the result from evaluation with the current rule set.

*Current Rule Set:* {current_rules}
*Sample Evaluation Results Based on Current Rules:* {error_analysis}

**Rule Refinement Guidelines.**
*Rule Refinement Process*

1. **Error Analysis and Tracing**:
    - Identify misjudged samples where the evaluation result is inconsistent with the manual label.
    - Analyze the reasoning process for each to determine which rules were applied and caused the misjudgment.
2. **Rule Diagnosis and Modification**:

- For incorrectly applied rules, check if their descriptions are ambiguous or their scope is inappropriate, and modify them to better fit the intended semantics.
- Delete rules that are inconsistent with the current semantics, cannot be generalized, or are misleading.
- Merge redundant or duplicate rules, refining their common logic to enhance their expressive power and applicability.

3. **Rule Augmentation**:
   - For new patterns exposed in the misjudged samples that are not covered by the existing rule system, add new rules to cover the corresponding judgment logic.

**Requirements for Each Rule.**

- Each rule must focus on specific structural and semantic differences in SQL, clearly reflecting the logical basis for an `equivalent` or `non-equivalent` judgment.
- Rules must be as detailed and specific as possible, and must **not** rely on execution results or result sets.
- Rules must be generalizable to SQL evaluations with similar structural or semantic features, not tied to a single specific sample.
- Examples may be used to illustrate the context in which a rule applies, but should focus on key SQL fragments rather than full queries.
- Rules that determine equivalence must be rigorously constructed to avoid false positives.

**Output Format.**

- First, provide your reasoning process step-by-step.
- Then output the final refined rules in strict JSON format:

```json
{
  "rules": [
    {
      "description": "A clear logical explanation of why a certain
  pattern leads to an equivalent/non-equivalent judgment.",
      "type": "True/False"  // True indicates support for
  equivalence; False indicates a judgment of non-equivalence.
    }
  ]
}
```

**Prompt Template: Rule Aggregation**

**Task Description.** We are building an LLM-based SQL equivalence evaluation system to determine whether a predicted SQL query (`pred_sql`) is *functionally equivalent* to the ground-truth SQL query (`gold_sql`).
The complete evaluation process consists of two major phases:

1. **Rule Extraction (Training)** – Derive evaluation rules from labeled samples to simulate the logic and criteria of manual evaluation (labels).
2. **Rule-Based Evaluation (Inference)** – Use the extracted rules to evaluate whether the predicted SQL is functionally equivalent to the ground-truth SQL.

The **Rule Extraction** phase is divided into two parts:

1. **Grouped Rule Extraction** – Since the number of samples can be large and difficult to process at once, they are grouped. Rules are then extracted from each group, resulting in local rules applicable to that specific group.

2. **Rule Aggregation** – These localized rules are then summarized and merged to form a comprehensive, general-purpose, and high-coverage evaluation rule set.

You are now responsible for the **Rule Aggregation** phase.

**Input.**

- **Rule Sets**: Several sets of local rules generated from clustering. Each rule includes:
  - `description` — the rule content
  - `type` — `True` (equivalent) or `False` (non-equivalent)
- Rule content overview: `{cluster_rules}`

**Rule Aggregation Guidelines.**
*Aggregation Goal*

- Systematically merge rules with similar semantics or judgment criteria to form a **global, general, and information-complete** rule set.

*Aggregation Process*

1. **Merge Similar Rules**
   - Compare semantic content and merge rules with similar judgment logic and application scope.
   - Retain all valuable judgment criteria and example patterns from merged rules.

2. **Handle Conflicts and Redundancy**
   - Prefer rules that are more general and broadly applicable.
   - Merge or remove duplicate or overly narrow rules.

**Requirements for Each Rule.**

- Focus on specific structural or semantic SQL differences.
- Rules must be based on **query structure only**, not execution results.
- Be clear, precise, and generalizable where possible.

**Output Format.**

- First provide your reasoning process step-by-step.
- Then output the final rules in strict JSON format:

```json
{
  "rules": [
    {
      "description": "Description of the merged rule.",
      "type": "True/False"
    }
  ]
}
```

**Prompt Template: Evaluation**

**Role.** You are an expert in SQL equivalence evaluation, focusing on assessing the **functional equivalence** between SQL queries. Your core task is to determine if a **Predicted SQL** is functionally equivalent to a **Ground-Truth SQL**.

*Note:* Functional equivalence means that, across all valid database states, both SQLs return the same result **and** achieve the same logical intent—even if their syntax differs.

**Evaluation Process.**

1. **Functional Equivalence Judgment**
   - Compare the Predicted SQL with the Ground-Truth SQL. Based on the evaluation rules, determine if the Predicted SQL is functionally equivalent to the Ground-Truth SQL.

2. **Return Final Decision**
   - Output a score between 0 and 1 to represent functional and logical consistency.
   - **Do not only use binary scores** like 0.00 or 1.00. Provide a **smooth and continuous score** to reflect the degree of functional equivalence.
   - A score closer to 1.00 indicates high equivalence; a score closer to 0.00 indicates low equivalence.
   - Scores $\geq 0.50$ are considered functionally equivalent; scores $< 0.50$ are not.
   - You must include a detailed reasoning to justify the score.

**Evaluation Rules.** Each rule includes:
   - `description` — describing the logic behind equivalence or non-equivalence.
   - `type` — `True` for equivalence, `False` for non-equivalence.

*Input Rule Set:* `{applicable_rules}`

**Input.**
   - **Predicted SQL:** `{predicted_sql}`
   - **Ground-Truth SQL:** `{reference_sql}`

**Output Format.**
   - The output must be a JSON block enclosed in ```json.
   - Fields include:
     - `"reasoning"`: A detailed explanation of the judgment process.
     - `"score"`: A float between `0.00` and `1.00`, with at least two decimal places.

**Output Format Example.**

```json
{
  "reasoning": "The reasoning process for the judgment.",
  "score": 0.87
}
```

### A.2.2 PROMPT TEMPLATES USED IN BOND-QA

**Prompt Template: Rule Extraction**

**Background.** We are building a **comprehensive SQL evaluation agent** that judges whether the predicted SQL can correctly answer the user's question. The full process consists of two steps:

1. **Rule Extraction (Training)** — Derive evaluation rules from labeled samples to simulate the reasoning logic and evaluation standards used by human annotators.

2. **Rule-Based Evaluation (Inference)** — Apply the extracted rules to assess whether each predicted SQL correctly answers the user's question.

You are responsible for the **Rule Extraction** stage.

**Input.**

- A cluster of semantically similar, labeled samples obtained via question clustering.
- Each sample contains:
  - User question
  - Predicted SQL
  - Reference SQL
  - Execution results of both SQLs
  - Human annotation (whether the predicted SQL is judged correct by human evaluation)
- The dataset is from the fixed-income domain, containing DBQA samples on bond-related data, all manually annotated.

**Your Task.**

- **Core Objective:**
  - Thoroughly analyze the samples to uncover the reasoning patterns behind the human "correct/incorrect" judgments.
  - Abstract a set of evaluation rules that are both **generalizable** and **operational**.
  - These rules will be used later to determine whether a predicted SQL truly satisfies the user's query intent.

**Principles for Rule Formulation.**

1. **Evaluation Dimension:** Focus on whether the predicted SQL can answer the user's question, ignoring non-critical differences such as SQL syntax, field order, or naming variations.
   - In practice, execution results are compared first; if they differ, rules are applied to determine whether the predicted SQL is still reasonable.
   - Therefore, rules should be grounded in **SQL structural logic**—selection logic, chosen fields, aggregation methods, and other structural features—to determine which differences are acceptable and which are not.

2. **Case Differentiation:** Clearly distinguish conditions under which a predicted SQL should be judged correct from those under which it should be judged incorrect.

3. **Domain Context:** Incorporate the dataset's specific domain context relevant to the question. Maintain transferability; avoid hard-coding specific table or field names.

4. **Rule Requirements:**
   - Rules must be clearly described, detailed, and specific.
   - They must be sufficiently general to apply to SQLs with similar structural or semantic characteristics beyond the originating sample.
   - A sample may be cited as an **illustrative example**, highlighting only the key aspects that support the rule (do not display full sample details or original sample IDs).

5. **Note:** The reference SQL is only a reference answer—it is not the only correct answer.

**Output Format Requirements.**

- Final output must be enclosed in " ```json.

- The top-level key must be `"rules"`, whose value is a list of rules (no limit on the number).
- Each rule must contain:
  - `"description"` — textual description of the rule (in the same language as the question in the sample).
  - `"type"` — evaluation logic indicator (`"True"` if this rule supports judging a predicted SQL as correct, `"False"` if it supports judging it as incorrect).

**Output Example.**

```json
{
  "rules": [
    {
      "description": "A clear logical explanation of why a certain
  pattern leads to an equivalent or non-equivalent judgment.",
      "type": "True/False" // True = equivalent, False = non-
  equivalent
    }
  ]
}
```

**Prompt Template: Rule Refinement**

**Background.** We are building a **comprehensive SQL evaluation agent** to determine whether a predicted SQL (`pred_sql`) generated by a database question answering (DBQA) system can correctly answer the user's question, even if it differs from the reference SQL (`ref_sql`) in both SQL structure and execution results.

The complete evaluation process consists of two main steps:

1. **Rule Extraction (Training)** — Derive evaluation rules from labeled samples to simulate the reasoning logic and evaluation standards used by human annotators.
2. **Rule-Based Evaluation (Inference)** — Apply the extracted rules to determine whether each predicted SQL correctly answers the user's question.

You are now in the **refinement iteration phase** of the Rule Extraction stage. Your task is to modify and improve the existing rules so that they can be correctly applied to the current set of samples. You will receive:

- A set of **current evaluation rules**.
- A set of **evaluation results** produced by applying these rules to the current samples, along with human evaluation labels for comparison.

Your goal is to analyze the misjudged samples, identify the causes of these errors, and update or supplement the rules so they better reflect human evaluation logic—ensuring that the revised rules work correctly on the current sample set.

**Input.**

- **Current Evaluation Rules**:

  {current_rules}

- **Evaluation Sample Analysis** (The "actual label" is the human evaluation result; the rest of the evaluation-related information comes from the model's evaluation results based on the current rules):

  {error_analysis}

**Rule Refinement Guidelines. Rule Refinement Process** The objective of refinement is to improve alignment between automated evaluation results and human judgments by modifying, adding, or removing rules so they more accurately capture the logic humans use to decide whether a predicted SQL correctly answers the question.

The process involves:

1. **Error Analysis & Root Cause Identification** Identify misjudged samples where the automated evaluation result differs from the human label. For each, analyze the evaluation reasoning process and determine which rules were applied and led to the misjudgment.

2. **Rule Diagnosis & Modification**
   - For incorrectly applied rules, check if the description is ambiguous or overly broad, and modify it to better reflect the intended semantics.
   - Remove rules that are inconsistent with the current semantics, cannot generalize, or may cause misleading judgments.
   - Merge redundant or duplicate rules, extracting common logic to improve clarity and generality.

3. **Rule Addition** If new patterns emerge from the misjudged samples that are not covered by the current rule set, add new rules to capture these patterns.

Through this process, iteratively optimize the rule set to make SQL equivalence evaluation more accurate and robust for the current sample set.

**Output Format.**

- First, present your reasoning process in plain text, showing your step-by-step thought process.
- The **final result** must be enclosed in "```json.
- The JSON top-level key must be `"rules"`, whose value is a list (no limit on number of rules).
- Each rule must have:
  - `"description"` — a clear logical statement describing the conditions under which the predicted SQL should be considered correct/incorrect.
  - `"type"` — either `"True"` (supports judging predicted SQL as correct) or `"False"` (supports judging it as incorrect).

**Example Output.**

```json
{
  "rules": [
    {
      "description": "Clear logical explanation of when a predicted
    SQL is correct/incorrect",
      "type": "True/False"  // True = supports correctness; False =
    supports incorrectness
    }
  ]
}
```

**Prompt Template: Rule Aggregation**

**Background.**
We are building a **comprehensive SQL evaluation agent** to assess whether a predicted SQL in a database question answering (DBQA) system can effectively answer the user's question. In the **Rule Extraction** phase, a set of *local evaluation rules* has been generated for each semantic cluster of samples. Your responsibility is the **Rule Aggregation** phase, where the goal is to merge multiple sets of local rules into a unified **global evaluation rule set** applicable to the entire dataset.

**Full Process Overview.**

1. **Rule Extraction (Training)** — Derive evaluation rules from labeled samples to simulate the reasoning logic and evaluation standards of human annotators.

2. **Rule-Based Evaluation (Inference)** — Apply the extracted rules to assess whether each predicted SQL correctly answers the user's question.

The **Rule Extraction** stage itself consists of two sub-steps:

1. **Cluster-wise Rule Extraction** — Since the dataset is large, it is divided into clusters. Rules are extracted for each cluster separately, producing local rules for that cluster.

2. **Rule Aggregation** — These local rules are then generalized, merged, and consolidated into a single global evaluation rule set with high coverage and broad applicability.

You are responsible for the **Rule Aggregation** step.

**Input.**

- **Rule Collections** (each rule contains a description and a type, where the description specifies the evaluation logic and the type indicates the evaluation direction):

    {cluster_rules}

**Rule Aggregation Guidelines. Aggregation Objective** Your goal is to systematically merge rules by integrating semantically similar or logically consistent rules into unified ones, resulting in a **global, general-purpose, and complete** evaluation rule set.
**Aggregation Process**

1. **Merge Similar Rules**

    - Compare the semantic meaning of different rules and merge those with similar judgment criteria or applicability.
    - When merging, **retain all valuable judgment conditions** and any illustrative examples from each rule, ensuring that no critical information is lost in the process.

2. **Handle Conflicts and Redundancy**

    - If conflicts or overlapping coverage are found:
        - Prioritize keeping rules that are more general, broader in scope, or easier to apply.
        - Remove or merge redundant, repetitive, or overly narrow rules.

**Output Format.**

- First, present your reasoning process (merging approach) in plain text, showing your step-by-step thought process.

- The **final result** must be enclosed in "```json.

- The JSON top-level key must be `"rules"`, whose value is a list (no limit on number of rules).

- Each rule must have:

    - `"description"` — a clear logical statement describing the conditions under which the predicted SQL should be considered correct/incorrect.

– `"type"` — either `"True"` (supports judging predicted SQL as correct) or `"False"` (supports judging it as incorrect).

**Example Output.**

```json
{
  "rules": [
    {
      "description": "Clear logical explanation of when a predicted
    SQL is correct/incorrect",
      "type": "True/False"  // True = supports correctness; False =
    supports incorrectness
    }
  ]
}
```

**Prompt Template: Evaluation**

**Role.** You are an SQL evaluation expert, skilled at assessing Text-to-SQL systems. Your task is to determine, based on the given *user question*, *predicted SQL and its execution result*, and *reference SQL and its execution result*, whether the predicted SQL's output can directly and correctly answer the user's question, or whether the correct answer can be inferred through reasoning.

**Evaluation Objective.** Judge whether the `predicted_sql` and its execution result can address the user's question directly, or whether the correct answer can be inferred from the SQL and its execution result. Prioritize checking whether the execution results match:

- If they match and satisfy the evaluation rules, return **True**.
- If they do not match, determine correctness based on the evaluation rules.

**Note:** The `reference_sql` is only a reference answer — it is not the only correct answer.

**Evaluation Rules.**

{applicable_rules}

**Input.**

- **User Question:**

  {question}

- **Current Time:**

  {current_time}

- **Predicted SQL:**

  {predicted_sql}

- **Predicted SQL Execution Result:**

  {predicted_result}

- **Reference SQL:**

  {reference_sql}

- **Reference SQL Execution Result:**

  {reference_result}

- **Background Knowledge:**

  {background_knowledge}

**Output Format Requirements.**
- Output must be enclosed in "```json.
- Do not include any additional explanatory text outside the JSON.

**Output Format.**

```json
{
  "reasoning": "Explanation of the judgment process",
  "result": true/false  // true = predicted SQL can correctly answer
    the question; false = it cannot
}
```

## A.3 CASE STUDY

In this section, we present illustrative examples from all three datasets to demonstrate how the introduction of rule guidance corrects evaluation outcomes. Each case highlights a situation where, without rules, the evaluator would misjudge the semantic correctness of a predicted SQL query. By applying our extracted rules, the evaluator is able to reason more faithfully about equivalence, leading to corrected decisions. These examples show concretely how our rules operate in practice.

### A.3.1 CASE STUDY OF BOND-QA

**NL Query**
How many secondary-market cash-bond trades and repo trades with **A** occurred in 2024? Report the counts separately.

**SQLs**
*Predicted*

```
SELECT
  CASE
    WHEN "trade_type" = 'Cash-Bond' THEN 'Cash-Bond'
    WHEN "trade_type" IN ('Buyout_Repo', 'Pledged_Repo') THEN 'Repo'
  END AS "trade_category",
  COUNT(DISTINCT "instruction_id") AS "trade_count"
FROM "trade_instructions"
WHERE "counterparty" = 'A'
  AND "trade_date" LIKE '2024%'
  AND "trade_status" = 'Confirmed'
  AND "trade_type" IN ('Cash-Bond', 'Buyout_Repo', 'Pledged_Repo')
  AND "market" IS NOT NULL
GROUP BY
  CASE
    WHEN "trade_type" = 'Cash-Bond' THEN 'Cash-Bond'
    WHEN "trade_type" IN ('Buyout_Repo', 'Pledged_Repo') THEN 'Repo'
  END;

-- Execution Result
```

```
-- trade_category | trade_count
-- Cash-Bond      | 1825
-- Repo           | 0
```

*Reference*

```sql
SELECT
  SUM(CASE WHEN "trade_type" = 'Cash-Bond' THEN 1 ELSE 0 END)
  AS "cash_bond_count",
  SUM(
    CASE
      WHEN "trade_type" IN ('Pledged_Repo', 'Buyout_Repo') THEN 1
      ELSE 0
    END
  ) AS "repo_count"
FROM "trade_instructions"
WHERE
  "source"       = 'Sales'
  AND "counterparty" = 'A'
  AND "trade_status" = 'Confirmed'
  AND "trade_date"  >= '2024-01-01'
  AND "trade_date"  <= '2024-12-31';

-- Execution Result
-- cash_bond_count | repo_count
--      1825        |      0
```

**Label:** 1

**Evaluation without Rules**

- **Result:** False

- **Reasoning:** The prediction returns only the cash-bond row when repo trades are zero, so the repo category does not appear explicitly. Interpreting the absence as zero is disallowed; therefore the answer is deemed incomplete.

**Evaluation with Rules**

- **Result:** True

- **Reasoning:** Core constraints align with the question: counterparty = A, time range = 2024, and status = confirmed. The reference adds a source filter (`source = 'Sales'`) not mandated by the question. Under grouping semantics, a missing category implies a zero count; thus the prediction directly yields the required counts (cash-bond count and repo = 0), satisfying the user request.

**Rule Triggered** When the predicted_sql and reference_sql differ in implementation details such as aggregation methods, grouping fields, filtering conditions, or field naming, but the execution result of the predicted SQL fully and accurately covers all the key information required by the user query (including fields, grouping categories, etc.), it should still be judged as correct. This applies even if certain categories are not explicitly shown in the result, as long as they can be logically inferred to be zero through standard SQL grouping and aggregation rules (e.g., when a category has no data and therefore does not appear in the output).

## A.3.2 CASE STUDY OF SPIDER-PAIR

**NL Query**

How many models does each car maker produce? List maker full name, id and the number.

**SQLs**

*Predicted*

```
SELECT fullname, id, COUNT(model) AS model_count
FROM model_list
LEFT JOIN car_makers ON model_list.maker = car_makers.id
GROUP BY fullname, id
```

*Reference*

```
SELECT t.FullName AS FULLNAME, t.Id AS ID, COUNT(*) AS EXPR$2
FROM (SELECT Id, FullName FROM car_makers) AS t
INNER JOIN (SELECT Maker FROM model_list) AS t0
ON t.Id = t0.Maker
GROUP BY t.Id, t.FullName
```

**Label:** 0

**Evaluation without Rules**

- **Result:** True ;(*Score:* 0.92)

- **Reasoning:** Both SQL queries aim to return, for each car maker, the maker's full name, id, and a count of associated models. However, there are notable differences in their join types and counting logic. The Ground-Truth SQL uses an INNER JOIN between car_makers and model_list, ensuring only makers with at least one model are included. It counts all joined rows (COUNT(*)), which corresponds to the number of models per maker. The Predicted SQL uses a LEFT JOIN from model_list to car_makers, grouping by fullname and id, and counts the number of models (COUNT(model)). This means it will include all makers from car_makers, even those with zero models, but since the join is from model_list, only makers present in mode_list will appear. However, the grouping and join direction in the Predicted SQL may result in missing makers with zero models, similar to the INNER JOIN in the Ground-Truth SQL. The main difference is that the Predicted SQL's COUNT(model) will not count NULLs, so if there are model_list rows with NULL model values, those won't be counted, whereas COUNT(*) in the Ground-Truth SQL would count all joined rows. Additionally, the column aliases differ, but this does not affect functional equivalence. Overall, the queries will return the same results in most practical cases, except for edge cases involving NULLs in the model column. Therefore, the functional equivalence is high but not perfect.

**Evaluation with Rules**

- **Result:** False ;(*Score:* 0.40)

- **Reasoning:** Both queries aim to return, for each car maker, the maker's full name, id, and a count of associated models. However, there are important differences in their logic and join types. The Predicted SQL uses a LEFT JOIN from model_list to car_makers, grouping by fullname and id, and counts the number of models per maker. The Ground-Truth SQL uses an INNER JOIN from car_makers to model_list, grouping by id and fullname, and counts the number of joined rows (COUNT(*)).

  Key differences: 1. Join Type: The Predicted SQL uses LEFT JOIN, which will include makers even if they have no models (with model_count = 0), while the Ground-Truth SQL uses INNER JOIN, which will only include makers that have at least one model. This can lead to different result sets if there are makers with no models. 2. COUNT Argument: The Predicted SQL counts model (COUNT(model)), which will count only non-null model values. The Ground-Truth SQL uses COUNT(*), which counts all joined rows. In this schema, if a model_list row exists, model is likely non-null, so this is probably equivalent, but not guaranteed unless model is non-nullable. 3. Grouping Columns: The grouping columns are the same in content (fullname/id vs. t.FullName/t.Id), just with different order, which is not significant unless order is required. 4. Projection: Both queries output the same logical columns (fullname, id, count of models).

The most significant difference is the join type: LEFT JOIN vs. INNER JOIN. This means the Predicted SQL will include makers with zero models, while the Ground-Truth SQL will not. This is a logical difference in the result set, violating rule 14 (join type changes row inclusion). Therefore, the queries are not functionally equivalent. However, in the case where all makers have at least one model, the results would match, so the queries are not completely unrelated. Thus, a score reflecting partial equivalence is appropriate.

**Rule Triggered**

If the set of columns in `SELECT`, `WHERE`, `GROUP BY`, or `JOIN` clauses differs between the queries, and these columns are not aliases or semantically equivalent, the queries are not equivalent.

### A.3.3 CASE STUDY OF NL2SQL-BUGS

**NL Query**

Which Russian set of cards contains the most cards overall?

**Candidate SQL**

```
SELECT T2.setCode
FROM sets AS T1
INNER JOIN set_translations AS T2
    ON T1.code = T2.setCode
WHERE T2.language = 'Russian'
ORDER BY T1.baseSetSize DESC
LIMIT 1
```

**Label:** 0

**Evaluation without Rules**

- **Result:** True
- **Reasoning:** The NL query asks for the Russian set of cards (i.e., set where there is a Russian translation) that contains the most cards overall. The schema shows that 'sets' has 'baseSetSize' (number of cards in the set), and 'set_translations' links 'setCode' to 'sets.code' and has language. The SQL joins sets and set_translations on 'setCode', filters for Russian, orders by 'baseSetSize' DESC, and limits to 1. However, the SELECT clause returns 'T2.setCode' (from set_translations), which is correct since setCode is the set identifier. The query does not use an aggregate function (MAX), but the ORDER BY ... DESC LIMIT 1 pattern is a valid way to get the set with the maximum baseSetSize. Therefore, the query is semantically correct and matches the NL query.

**Evaluation with Rules**

- **Result:** False
- **Reasoning:** The NL query asks for the Russian set of cards (i.e., set where there is a Russian translation) that contains the most cards overall. The schema indicates that 'sets.baseSetSize' gives the number of cards in a set, and 'set_translations.language' specifies the language. The SQL joins 'sets' and 'set_translations' on set code, filters for Russian, orders by 'baseSetSize' descending, and limits to 1. However, the SELECT clause returns 'T2.setCode', which is the set code, not the set name. The NL query asks for the "set", which is ambiguous but usually refers to the set name ('sets.name'), not the code. Therefore, the query returns the set code, not the set name, which is an attribute mismatch.

**Rule Triggered** (4) The SELECT clause omits required fields, includes fields not requested by the question (when this changes the meaning or violates explicit requirements), or misformats the output.

**All Rules** A SQL query is judged incorrect (False) if ANY of the following are true: (1) It omits or misapplies any required entity, attribute, constraint, filter, or join from the natural language question, or maps them to the wrong schema columns or values (including case or value mismatches); (2) It uses incorrect, missing, or extraneous joins, filters, columns, or logic that would change the result set or meaning, including joining on non-key or unrelated columns, or failing to join necessary tables; (3) It applies aggregations, calculations, or groupings incorrectly, including using the wrong function, missing DISTINCT when required, failing to handle ties or uniqueness, or returning results at the wrong granularity; (4) The SELECT clause omits required fields, includes fields not requested by the question (when this changes the meaning or violates explicit requirements), or misformats the output (e.g., missing both members of a pair, not using YES/NO when required, or not matching required output format); (5) The query fails to map natural language terms to the correct schema fields or values, including incorrect handling of date formats, string case, or value sets; (6) The query is not robust to all plausible database states, such that for some valid data it would return an incorrect, incomplete, or ambiguous answer (e.g., assuming uniqueness where it may not exist, or using '=' instead of 'IN' for subqueries that may return multiple rows); (7) The query contains SQL syntax errors or uses hardcoded IDs or values not referenced in the question; (8) Any error or omission that could cause the result to be wrong, incomplete, ambiguous, or not fully answer the question, including partial answers to multipart questions, incorrect logic for counting, grouping, or aggregating, or output in the wrong format when the question explicitly requires a specific format.

**Error Type**

- Attribute-Related Errors: Attribute Mismatch

## A.4 RULES EXAMPLE

Below are some representative rules extracted from the Spider dataset. As we can see, these rules align well with human intuition. Green-highlighted rules indicate cases that are judged as **equivalent**, while red-highlighted rules represent cases that are judged as **non-equivalent**

---

**Examples of Semantic Equivalence Rules**

**[Rule 1]** Differences in the use of `GROUP BY` and `DISTINCT` for deduplication are considered equivalent if the deduplication logic is preserved and no aggregation is present. For example, `SELECT DISTINCT column` vs `SELECT column GROUP BY column` are equivalent. However, if deduplication is omitted when duplicates are possible, the queries are non-equivalent.

**[Rule 2]** Differences in the use of subqueries in the `FROM` clause, derived tables, or wrapping a table/column selection in a subquery (e.g., `FROM (SELECT ...)` vs direct table reference) do not affect equivalence if the subquery does not filter, transform, or otherwise alter the data, and the logical meaning and output columns remain unchanged. Additional subquery nesting or aliasing is also equivalent if the logical operation and selected columns are the same.

**[Rule 3]** Differences in the use of aggregation functions (e.g., `COUNT(*)`, `COUNT(1)`, `COUNT(column)`) are considered equivalent if the aggregation semantics are preserved, the grouping keys are the same, and the counted column is either NOT NULL or a key. However, `COUNT(DISTINCT column)` and `COUNT(*)` are only equivalent if the column is unique and non-null in context.

**[Rule 4]** If the predicted SQL uses a different aggregation function, operates on a different column, or applies aggregation at a different grouping level than the ground-truth SQL, the queries are non-equivalent. This includes differences in `COUNT(*)`, `COUNT(column)`,

COUNT(DISTINCT ...), SUM, MAX, MIN, or grouping columns, unless the extra grouping columns are functionally dependent or redundant.

**[Rule 5]** If the predicted SQL uses different ORDER BY or LIMIT clauses, or omits them when present in the ground-truth SQL, and this affects which rows are returned or if the order is required by the question, the queries are non-equivalent. For example, ORDER BY rating DESC LIMIT 1 vs omitting LIMIT or changing the sorting order.

**[Rule 6]** If the predicted SQL uses subqueries or aliases in a way that changes logical relationships, data scope, or filtering compared to the ground-truth SQL, the queries are non-equivalent. This includes differences in subquery placement (e.g., WHERE vs JOIN), correlated subqueries, or logic that alters the rows or values being considered.

## A.5 USE OF LARGE LANGUAGE MODELS

We only used large language models to polish the writing of this paper, and did not use them in any other part of the research.

