# OpenReview forum: "When the Gold Answer Isn't the Only Right One: Evaluating Database QA via LLM-Induced Rule Guidance"
_ICLR.cc/2026/Conference — ICLR 2026 Conference Withdrawn Submission_

### Official Review · Reviewer_uArN · 2025-10-30

**Soundness:** 2
**Presentation:** 2
**Contribution:** 2
**Rating:** 2
**Confidence:** 4

**Summary:**

This paper addresses a fundamental limitation in evaluating Text-to-SQL (NL2SQL) systems: traditional metrics like Execution Accuracy (EX) treat the reference SQL as the only correct answer, which fails in real-world cases where multiple semantically correct SQLs can exist for a single question. To tackle this, this paper proposes HLSemEval, which uses LLMs to derive evaluation rules from annotated data and these rules are integrated into LLM prompts at inference time to improve semantic evaluation. The evaluation shows consistent (but marginal) F1 score improvement over baseline methods.

**Strengths:**

1. It highlights a common issue where a generated SQL query can be correct in terms of user intent but still produce a different result from the reference answer. This can happen when the question is ambiguous, there are multiple valid ways to write the query, the query results are presented in different ways or the data has inconsistencies. I've seen this happen in practice too, so the problem is very relevant.

2. The method is clearly explained and easy to understand.

3. The experiments are through in that the authors evaluate across several datasets and LLMs, compare against relevant baselines, and include ablation studies.

**Weaknesses:**

1. Lack of Formal Definitions and Early Example: The notions of semantic intent and semantic correctness are central to the paper but are only informally described. A more formal definition should be introduced earlier to reduce ambiguity. The concept of "rules" is also vague; providing concrete examples up front would help ground the reader’s understanding of how these rules operate and how they differ from existing evaluation strategies. (I noticed that there are examples in the appendix but those should come earlier in the paper.

2. Potential Bias from Annotated Data: Since the proposed framework derives evaluation rules from human-labeled examples, it risks inheriting biases and errors from the annotations themselves. This raises concerns about whether the method offers substantial benefits over simpler strategies like EX matching or prompting LLMs directly with carefully crafted instructions.

3. Rules are dataset-specific: The authors note that extracted rules tend to reflect dataset-specific task settings (e.g., Bond-QA vs Spider-Pair). While this shows adaptability, it also raises concerns about generalization. If rules must be re-learned per dataset (and per domain), the scalability and portability of the approach may be limited in practice. Moreover, the existence of different rule sets for different datasets implies that the underlying definition of semantic correctness is not consistent across datasets. This raises the question of whether the framework is truly capturing a universal notion of semantic correctness or merely reproducing dataset-specific evaluation heuristics.

4. Lack of Transparency on the BondQA dataset: The paper states that the Bond-QA dataset consists of 900 real-world samples annotated by “in-house experts” for semantic correctness. However, it does not explain the annotation process in detail: How many annotators were involved? How do they evaluate semantic correctness? Are those labels reliable?

5. Marginal Gains vs. Supervision Cost: Although the rule-guided approach improves F1 scores on some benchmarks, the gains (e.g., at most +2.16 on Bond-QA) are relatively modest (Table 1). It's unclear whether these improvements justify the cost of rule induction, clustering, and human annotation, especially when compared to high-performing baselines that use minimal supervision or prompt-only setups.

**Questions:**

See weaknesses, particularly

1. How Bond-QA dataset is annotated? Did "in-house experts" follow certain guidelines to label semantic correctness? Any quality control on that?

2. How do you justify that the notation of "semantic correctness" needs to be re-learned per dataset? The notation of "semantic correctness" should be general across questions from different datasets, right?

3. How do you justify the cost of rule induction, clustering, and human annotation given the performance gain is modest?

---

### Official Review · Reviewer_5wyn · 2025-10-31

**Soundness:** 3
**Presentation:** 3
**Contribution:** 3
**Rating:** 8
**Confidence:** 3

**Summary:**

In this paper, the authors propose HLSemEval, an LLM-based evaluation framework for NL2SQL tasks and aim to address some of the limitations of traditional Execution Accuracy (EX) metrics.
They argue that EX fails to recognize semantically correct SQL queries that produce different execution results from the reference. Here the semantically correct refers to the perception of the end user.
As a result, the authors design a rule generation enhancing framework that learns interpretable evaluation rules from annotated data instead of updating model parameters.
The framework consists of three stages 1) clustering, 2) rule extraction, and 3) self-evaluation on the labeled data points. Experiments on Bond-QA, Spider-Pair, and NL2SQL-BUGs show that rule-guided evaluators align more closely with human judgments and outperform existing methods such as EX, FLEX, and Miniature & Mull. Also, the ablation studies confirm the importance of the clustering and rule consolidation steps for robustness and interpretability.

**Strengths:**

S1. The paper introduces a novel and interpretable rule-based evaluation framework (HLSemEval) that effectively addresses the long-standing issue with the traditional execution-based metrics.

S2. The rules are automatically identified from the labeled example. The approach is sound and tailored toward user's intent.

S3. Assuming the Bond-QA dataset will be publicly released, this paper will provide a valuable benchmark for evaluating pragmatic correctness in NL2SQL, offering a practical testbed for future research on human-like evaluation beyond execution accuracy.

**Weaknesses:**

W1. The framework relies on human-annotated data for rule induction, which may limit scalability to new domains or unseen database schemas without additional labeling effort. Perhaps a more user friendly annotation framework need to be introduced, such as pairwise preferences.

W2. I would prefer the authors to look at additional dataset for functional (semantic) equivalence beyond Spider-Pair. More specifically, please add experiment results on DB query rewrite datasets (known for their correctness). For example, the Calcite test suites adopted in LLM SQL Solver and can be found at https://github.com/SJTU-IPADS/SQLSolver.

**Questions:**

Please see W1-W2.

---

### Official Review · Reviewer_QdzR · 2025-11-03

**Soundness:** 2
**Presentation:** 3
**Contribution:** 2
**Rating:** 2
**Confidence:** 5

**Summary:**

This paper studies a key limitation in current evaluation pipelines for text-to-SQL: existing benchmarks typically assume a single “gold” SQL query per question, even though many semantically equivalent SQL queries exist. The authors introduce a framework for supporting multi-correct outputs in NL2SQL evaluation, which identifies and canonicalizes correct alternatives through execution checks and semantic equivalence. Experimental studies show that substantial discrepancy between actual model correctness and accuracy under single-reference scoring.

**Strengths:**

S1. Semantic equivalence of SQL queries is an important problem for NL2SQL and beyond in general. Moreover, a natural language query may map to several SQL expressions or execution outputs that faithfully reflect the same intent.

S2. The authors propose a practical evaluation pipeline leveraging LLM extracted rules to predict the query correctness.

**Weaknesses:**

W1. The equivalence validation pipeline relies heavily on LLMs for query equivalence, which may miss nuanced cases (e.g., floating-point precision, NULL handling, nondeterministic functions) from SQL executions.

W2. The technical novelty or contributions are limited. HLSemEval relies on both high-quality human-annotations and LLMs to extract rules, which can be expensive in terms of adapting to different databases or datasets. Also it does not consider the ambiguities from NLQs and data. Hence it is not clear to me whether the proposed solution is practical in real-life applcations.

W3. The paper would benefit from a stronger experimental evaluation. Specifically, more baselines (either using LLM-as-a-judge, query execution results, or discovering latent user preference) should be compared against. Also additional ablation study on how many candidate queries are needed to reliably capture equivalence diversity is needed.

W4. The paper does not propose new equivalence or matching criteria. SQL execution can introduce high latency. The paper does not consider SQL logical/physical plans, which are used in other baselines already.

**Questions:**

Q1. How do you handle non-deterministic functions, floating-point tolerance, ordering, and NULL semantics when declaring equivalence?

Q2. What percentage of "equivalent" queries identified via execution turn out to be non-equivalent under relational algebra semantics?

Q3. How scalable is your approach to complex schemas or very large enterprise databases where execution cost is high?

Q4. How do you guarantee diversity of alternative programs rather than surface-level variants?

Q5. How much of the effect is due to benchmark artifacts vs. fundamental semantic ambiguity?

---

### Official Review · Reviewer_4atB · 2025-11-04

**Soundness:** 2
**Presentation:** 3
**Contribution:** 2
**Rating:** 4
**Confidence:** 3

**Summary:**

The paper argues that execution accuracy (EX) and match-based metrics are insufficient for evaluating NL2SQL in the LLM era: (a) different SQLs may be semantically correct despite differing results; (b) identical results can mask functional differences. It proposes HLSemEval, a two-stage framework that learns human-like, dataset-specific natural-language rules via clustering, cluster-wise rule extraction and refinement, and global rule aggregation, then uses those rules to guide an LLM evaluator at inference. On Bond-QA (semantic correctness with differing results), Spider-Pair (functional equivalence without execution), and NL2SQL-BUGs (no gold SQL), HLSemEval improves over LLM evaluators without rules, approaches a specialized graph-based evaluator (FuncEvalGMN) with far less supervision on Spider-Pair, and yields consistent gains across models, with case studies and ablations supporting design choices.

**Strengths:**

S1. Addresses real shortcomings of EX in multi-answer settings and false positives with coincidental equivalence.
S2. Interpretable rule library; explicit, reusable prompt templates; concrete case studies.
S3. Consistent gains across datasets and models; near parity with FuncEvalGMN using <10% training data on Spider-Pair.
S4. Evidence of rule transfer from stronger to weaker models.
S5. Covers three evaluation regimes (different results acceptable; same results but not equivalent; no gold SQL).

**Weaknesses:**

W1. Incremental novelty vs FLEX/LLM-SQL-Solver (LLM-as-judge with structured prompts) and vs Test-Suite Accuracy/FuncEvalGMN (functional semantics without EX). The proposed pipeline reads as a systematic engineering of existing rubric-guided evaluation rather than a fundamentally different evaluation principle. A head-to-head against the latest variants and stronger prompt/rubric baselines (with careful tuning) would better quantify incremental gains and clarify originality.

W2. Rule-learning depends on labeled data; unclear cross-domain generalization; possible encoding of dataset-specific biases.
Without evidence of transfer beyond the studied domains, rules risk overfitting to annotation conventions and schema idiosyncrasies. Please add label-noise sensitivity and cross-dataset transfer studies (train on A, test on B) to demonstrate robustness.

W3. Limited robustness analysis (adversarial/gaming), limited human-alignment verification (e.g., IAA, blinded studies). Evaluators can often be gamed by superficial rewrites or prompt artifacts; adversarial and counterfactual stress tests are needed. Provide inter-annotator agreement and blinded human-vs-model judgment comparisons to substantiate alignment claims.

W4. Cost/latency and reproducibility of clustering → multi-round extraction/refinement → aggregation are under-specified. Report end-to-end token usage, wall-clock, and model costs, and compare to Test-Suite and FuncEvalGMN to justify practicality. Release rule libraries, prompt templates, seeds, and scripts to enable exact reproduction of the reported numbers.

W5. Some baselines not re-run; potential version/fairness concerns acknowledged but unresolved. Re-executing baselines with matched model versions, decoding settings, and supervision budgets would reduce confounds. Include ablations with stronger/equally capable LLM evaluators and ensembles to ensure comparisons are balanced.

**Questions:**

1. How do choices of embedding model, clustering method, number/size of clusters affect performance and rule diversity? Please include sensitivity curves and rule overlap/coverage statistics.

2. Provide token counts, wall-clock, and model usage for rule extraction, refinement, aggregation, and inference with/without rules.

3. Evaluate resistance to superficial transformations and adversarial contrast sets; report error taxonomy and common failure modes.

4. Report inter-annotator agreement and a blinded study comparing evaluator decisions to human judgments; analyze disagreements.

5. How well do Spider-derived rules transfer to BIRD/other real schemas without retraining? Include cross-domain transfer experiments and detailed breakdowns.

---

### Note · Authors · 2025-12-02

I have read and agree with the venue's withdrawal policy on behalf of myself and my co-authors.